# Minimal Change Disease Is Associated with Mitochondrial Injury and STING Pathway Activation

**DOI:** 10.3390/jcm11030577

**Published:** 2022-01-24

**Authors:** Byung Chul Yu, Ahrim Moon, Kyung Ho Lee, Young Seung Oh, Moo Yong Park, Soo Jeong Choi, Jin Kuk Kim

**Affiliations:** 1Division of Nephrology, Department of Internal Medicine, Soonchunhyang University Bucheon Hospital, 170 Jomaru-ro, Bucheon 14584, Korea; nephroybc@schmc.ac.kr (B.C.Y.); futurelkh@schmc.ac.kr (K.H.L.); 108254@schmc.ac.kr (Y.S.O.); mypark@schmc.ac.kr (M.Y.P.); crystal@schmc.ac.kr (S.J.C.); 2Department of Pathology, Soonchunhyang University Bucheon Hospital, 170 Jomaru-ro, Bucheon 14584, Korea; armoon@schmc.ac.kr

**Keywords:** minimal change disease, mitochondrial injury, prognostic marker, stimulator of interferon genes pathway, urinary mitochondrial DNA

## Abstract

We hypothesized that minimal change disease (MCD) pathogenesis may be associated with mitochondrial injury, and that the degree of mitochondrial injury at the time of diagnosis may serve as a valuable prognostic marker. We compared urinary mitochondrial DNA (mtDNA) at the time of diagnosis in patients with MCD and age- and sex-matched healthy controls (MHC) (*n* = 10 each). We analyzed the site and signal intensity of immunohistochemical (IHC) staining of stimulator of interferon genes (STING) using kidney tissues at the time of diagnosis in patients with MCD. Patients with MCD were divided into high (*n* = 6) and low-intensity (*n* = 14) subgroups according to the signal intensity. Urinary mtDNA levels were elevated in the MCD groups more than in the MHC group (*p* < 0.001). Time-averaged proteinuria and frequency of relapses during the follow-up period were higher in the high-intensity than in the low-intensity subgroup (1.18 ± 0.54 vs. 0.57 ± 0.45 g/day, *p* = 0.022; and 0.72 ± 0.60 vs. 0.09 ± 0.22 episodes/year, *p* = 0.022, respectively). Mitochondrial injury may be associated with MCD pathogenesis, and the signal intensity of STING IHC staining at the time of diagnosis could be used as a valuable prognostic marker in MCD.

## 1. Introduction

Minimal change disease (MCD) is the cause of nephrotic syndrome in approximately 10% to 15% of adults [1], and is defined by the absence of histologic glomerular abnormalities, other than the fusion of podocyte foot process on ultrastructural analysis using electronic microscopy [2].

T-cell dysregulation has been proposed as a major contributor to podocytopathy [3,4,5], and the effectiveness of B-cell depletion therapy suggests B-cells role as drivers of disease [6,7]; however, MCD pathogenesis remains unknown. Although corticosteroid (CS) treatment is the first-line treatment for patients with MCD and is often effective [8], the mechanism is unclear, and response patterns vary between patients. The overall steroid responsiveness rate approached 70% to 90% after 12 weeks of treatment [9,10,11]. Of the responders, 25% to 35% never relapsed and 20% to 40% frequently relapsed [10,12,13]. So far, there are no factors that could predict treatment outcomes. Since the current treatment for MCD is suboptimal, pathogenesis clarification and the development of biomarkers that can predict the treatment response are needed.

The kidney is an organ with high energy demand and has abundant mitochondria. Mitochondrial dysfunction is associated with the pathogenesis of several kidney diseases [14,15,16,17,18,19]. Various acquired renal pathophysiological insults, including oxidative stress, renin-aldosterone-angiotensin system activation, and ischemia/hypoxia induce mitochondrial dysfunction, which induces podocyte injury, tubular cell damage, and endothelial dysfunction [20]. Mitochondrial damage leads to mitochondrial DNA (mtDNA) fragmentation and it is released into the cytosol. Cytosolic mtDNA activates the cyclic GMP-AMP synthase (cGAS)-stimulator of interferon genes (STING) pathway, resulting in kidney damage [21]. Recent animal studies have demonstrated that mitochondrial damage induces kidney inflammation and fibrosis via the cGAS-STING pathway using STING immunohistochemical (IHC) staining [19,21]. Additionally, mtDNA cytosolic fragments enter the systemic circulation and are released into the urine, a surrogate marker of mitochondrial injury in various kidney diseases [17,18,22].

We hypothesized that the MCD pathogenesis may be associated with mitochondrial injury, and that STING staining signal intensity in the kidney tissue of patients with MCD at diagnosis may be a valuable biomarker for predicting treatment outcomes. We examined whether urinary mtDNA was elevated in patients with MCD compared with age- and sex-matched patients with IgAN and healthy controls (MHC). We also compared the STING IHC staining and signal intensity between patients with MCD and age- and sex-matched IgAN patients, and assessed the correlation with STING IHC staining signal intensity and treatment outcomes in patients with MCD.

## 2. Materials and Methods

### 2.1. Study Population

This was a combined analysis of a prospective multicenter and retrospective single-center cohort study. To evaluate mitochondrial injury using urinary mtDNA copy numbers, we prospectively enrolled patients with MCD, age- and sex-matched healthy controls (MHC) and patients with IgAN as controls from the Cohort for Biomarker Inquiry of Renal Aggravation (COBRA) cohort, which collects clinical information, kidney tissue, serum, and 24 h-collected urine samples from patients with glomerulonephritis at Soonchunhyang University Seoul, Bucheon, and Cheonan Hospitals from May 2019 to December 2020 (*n* = 10 each). MHC were included if they had no history of specific diseases, drugs, or any abnormalities on health checkups at the time of enrollment. All participants provided written informed consent.

To evaluate the characteristics of STING IHC staining in patients with MCD, we retrospectively screened patients who had undergone kidney biopsy at Soonchunhyang University Bucheon Hospital between January 2001 and December 2018. Of the 735 patients who underwent kidney biopsy, 43 (5.9%) were diagnosed with MCD. We excluded 12 patients who were followed for <3 years due to follow-up loss and 11 patients whose pathology showed combined diseases, including IgAN, diabetic nephropathy, and hypertensive nephrosclerosis (*n* = 7), or difficulty in differentiating them from focal segmental glomerulosclerosis (FSGS; *n* = 4) (Figure 1). Finally, 20 patients with MCD and age-/sex-matched patients with biopsy-proven pure IgAN as controls were retrospectively enrolled.

Since IgAN is the most common primary GN worldwide, and the only known GN in which mitochondrial injury is related to its pathogenesis and clinical outcomes in a previous study [23], it was utilized as a control in this study.

### 2.2. Clinical and Laboratory Data Collection

We collected data on demographics and comorbidities, including a history of diabetes mellitus and hypertension. We obtained the body mass index and mean arterial pressure at the time of kidney biopsy. The type, duration, and total dose of administered medications, including immunosuppressive agents, angiotensin-converting enzyme (ACE) inhibitors, and angiotensin II receptor blockers (ARBs) during the follow-up period were reviewed. We obtained laboratory data at every follow-up visit. The eGFR was determined from serum creatinine values using the Chronic Kidney Disease Epidemiology Collaboration equation [24]. Proteinuria levels were obtained by 24-h urine collection, and time-averaged proteinuria was calculated as the average of the mean of proteinuria measurements every six months for each patient.

### 2.3. Urinary mtDNA Copy Number Quantification

Urinary copy numbers of the mtDNA genes nicotinamide adenine dinucleotide dehydrogenase subunit-1 (ND1) and cytochrome C oxidase subunit III (COX3) were measured by quantitative real-time polymerase chain reaction (qPCR) using 24-h urine collection samples in MHC from MHC and prospectively enrolled patients with MCD and IgAN at the time of kidney biopsy.

DNA was isolated from urine samples (1.75 mL) using DNA isolation kits (Norgen Biotek, Thorold, ON, Canada; cat. no. 18100). DNA concentrations were analyzed using a NanoDrop spectrophotometer (Thermo Fisher Scientific, Waltham, MA, USA). We performed qPCR using ND1 primers (forward 5′-AGTCACCCTAGCCATCATTCTACT-3′ and reverse 5′-GGAGTAATCAGAGGTGTTCTTGTGT-3′) and COX3 primers (forward 5′-AGGCATCACCCCGCTAAATC-3′ and reverse 5ʹ-GGTGAGCTCAGGTGATTGATACTC-3ʹ) (Thermo Fisher Scientific, Waltham, MA, USA), and 20 ng of template DNA/sample. The PCR conditions were as follows: 95 °C for 10 min, 40 cycles of 95 °C for 15 s, and 60 °C for 60 s. We corrected mtDNA copy numbers to the nuclear control gene, RNAse-P (nDNA; Thermo Fisher Scientific, Waltham, MA, USA; cat. no. 4403326) using human genomic DNA for the standard curve and calculated mtDNA copy numbers using Copy Caller software version 2.0 (Thermo Fisher Scientific, Waltham, MA, USA) and expressed it as an mtDNA/nDNA ratio [23,25].

### 2.4. Immunohistochemical Detection of STING

We analyzed the STING IHC stain and signal intensity in the kidney tissue of retrospectively enrolled MCD and matched IgAN patients. Kidney tissue was fixed with Mildform 10N (FUJIFILM Wako Pure Chemical Corporation, Osaka, Japan), and sections cut at a thickness of 3 mm were incubated with the appropriate primary antibody and processed with the Vectastain Elite ABC HRP kit (Vector Laboratories, Burlingame, CA, USA). The signal was detected by incubation with an ImmPACT diaminobenzidine peroxidase substrate (Vector Laboratories). The IHC stain site was evaluated for the entire kidney tissue, including the glomerulus and tubulointerstitium, and signal intensity was categorized from negative to 3+ by a single pathologist. Retrospectively collected patients with MCD were divided into high and low-intensity subgroups according to a signal intensity of STING IHC staining in glomeruli higher or lower than 2+, respectively.

### 2.5. Clinical Outcome Measures

In a retrospective study to assess the correlation between STING IHC staining signal intensity and treatment outcomes in patients with MCD, we analyzed within-subgroup differences in the mean annual rate of eGFR decline, rate of complete remission (CR) or partial remission (PR), time to remission, rate of relapse, and time to first relapse. CR was defined as proteinuria < 0.3 g per 24 h and PR defined as proteinuria > 0.3 g but <3.5 g per 24 h or a decrease in proteinuria by at least 50% from the baseline value at the time of kidney biopsy and <3.5 g per 24 h. A relapse was defined as proteinuria > 3.5 g per 24 h after CR or PR, according to the Kidney Disease Improving Global Outcomes (KDIGO) guidelines [26].

### 2.6. Statistical Analysis

Descriptive characteristics of the study population are reported as means ± standard deviations for continuous variables and as frequency counts with percentages for categorical and binary variables. The differences between the two groups were compared via Mann–Whitney and Wilcoxon signed-rank tests for continuous variables and either χ^2^ tests or Fisher’s exact tests for categorical variables, as appropriate. One-way ANOVA or the Kruskal–Wallis test for continuous variables and the chi-square test or Fisher’s exact test for categorical variables, as appropriate analysis methods, were used to compare the differences between the three groups. After one-way ANOVA, Bonferroni correction was used for post-hoc analysis. Urinary mtDNA copy numbers were log-transformed, and differences between groups were compared using the Mann-Whitney test. All statistical tests were two-sided, and *p*-values of less than 0.05 were considered statistically significant. All analyses were performed using SPSS version 25.0 for Windows (SPSS Inc., Chicago, IL, USA) or Graphpad Prism5 (GraphPad Software, La Jolla, CA, USA).

## 3. Results

### 3.1. Study Population

When we prospectively evaluated mitochondrial injury using urinary mtDNA copy numbers, we observed that proteinuria levels were higher in the MCD group than in the IgAN group. Both the MCD and IgAN groups showed higher proteinuria levels than the MHC group. More patients in the MCD and IgAN group were hypertensive and treated with ARBs or ACE inhibitors than in the MHC group. There were no significant differences in other baseline characteristics between the MCD, IgAN, and MHC groups (Table 1).

When we retrospectively studied the characteristics of STING IHC staining, we observed that baseline eGFR and proteinuria were higher in the MCD than in the IgAN group, but other baseline characteristics did not differ between the groups (Appendix A).

Among the enrolled patients, all patients with MCD and IgAN who were treated with corticosteroids were treated after the final diagnosis via kidney biopsy.

### 3.2. Urinary mtDNA Copy Numbers Comparison in the Three Groups in the Prospective Study

Log_10_ND1/nDNA and log_10_COX3/nDNA were significantly higher in the MCD and IgAN groups than in the MHC group. Log_10_ND1/nDNA and log_10_COX3/nDNA tended to be higher in the IgAN group than in the MCD group, but there was no significant difference between the groups (Figure 2).

### 3.3. STING IHC Stain Comparison in the MCD and IgAN Groups in the Retrospective Study

In patients with MCD, kidney tissue from 13 patients showed positive STING IHC staining, and the glomerulus was exclusively stained (Table 2). Both the endothelium and the epithelium of the glomerulus were stained. The IHC staining degree of the 13 patients with MCD showed various signal intensities from grades 0 to 3 (Figure 3A–D).

In patients with IgAN, kidney tissue from all 20 patients showed positive STING IHC staining. Various kidney structures were stained, and the staining sites varied between patients. Of the 20 total kidney tissue samples obtained from patients with IgAN, 13 were stained in the glomerulus, 18 in the tubule, and 17 in the interstitium (Table 2). In patients with IgAN, not only the endothelium but also the mesangium were stained. Some of the tubules were partially stained, and atrophic tubules were strongly stained. The endothelium and peritubular capillaries of the vessels were strongly stained (Figure 3E).

### 3.4. Treatment Response Analysis Based on the STING IHC Stain Signal Intensity in Patients with MCD

A total of 20 retrospectively enrolled patients with MCD were divided into high (*n* = 6) and low-intensity (*n* = 14) subgroups according to a signal intensity of STING IHC staining in glomeruli higher or lower than 2+, respectively. There was no significant difference in baseline characteristics, including eGFR and proteinuria, between the subgroups (Table 3). The proportion of patients treated with CS monotherapy or combined therapy of CS and other immunosuppressive agents including cyclophosphamide and cyclosporine A was not significantly different between the subgroups. Immunosuppressive treatment duration during the follow-up period was higher in the high-intensity subgroup than in the low-intensity subgroup. The duration and cumulative dose of CS administration tended to be higher in the high-intensity subgroup than in the low-intensity subgroup, but the difference was not statistically significant (Appendix A).

Time-averaged proteinuria and frequency of relapses during the follow-up period were higher in the high-intensity subgroup than in the low-intensity subgroup. Other outcome indices, including the mean annual rate of eGFR decline, time to CR or PR after treatment, and first relapse after CR or PR were not different between the subgroups (Table 4).

## 4. Discussion

To the best of our knowledge, this is the first study to investigate the role of mitochondrial injury in MCD pathogenesis and STING pathway activation in GN. Since urinary mtDNA levels are one of the most validated surrogate markers for mitochondrial injury in the kidney, and its clinical usefulness has been established in various kidney diseases [17,18,22,23,27], we used it to confirm that mitochondrial injury is involved in the MCD pathogenesis. Recently, we have conducted studies using urinary mtDNA for mitochondrial injury in GN pathogenesis. In a study on IgAN, urinary mtDNA levels were elevated in patients with IgAN, and changes in urinary mtDNA before and after treatment were inversely correlated with changes in eGFR [23]. It is not yet clear whether mitochondrial damage plays a role in GN pathogenesis other than IgAN. Our findings show that urinary mtDNA is also increased in MCD, which has a different pathogenesis from IgAN. Therefore, our findings suggest that mitochondrial injury is more likely to be a ubiquitous finding of GN than a unique finding of IgAN. 

Treatment guidelines for various GN have been developed; however, treatment results are still suboptimal [28]. If the mitochondrial injury is ubiquitous in GN pathogenesis, the treatment for mitochondrial injury could be an attractive complementary treatment option for patients with GN. To clarify this, further studies on other types of GN are needed. Recent preclinical studies using experimental models have shown that several mitochondria-targeted therapeutic approaches, including Mito-TEMPO [29,30], Mitoquinone [14,31], and Szeto-Schiller-31 peptide [32,33] have promising effects in various kidney diseases. These mitochondrial-targeted antioxidants suppress the production of mitochondrial-derived reactive oxygen species and ameliorate kidney damage and inflammation [34]. If the effectiveness of these drugs is verified in humans in the future, such a change in treatment paradigm is possible.

The major limitation of several studies that investigated mitochondrial injury using urinary mtDNA in various kidney diseases was that the mitochondrial injury site could not be identified [17,18,22,23,27]. In these studies, the correlation with prognosis was analyzed under the premise that mitochondrial injury degree and urinary mtDNA levels would have a positive correlation, but there was no way to confirm this premise. In this study, focusing on the results of recent studies that demonstrated that the STING pathway is activated by mitochondrial injury in the kidney [19,21,35], the mitochondrial injury site and the injury degree were analyzed through STING IHC staining. Interestingly, the staining was limited to the glomerulus in patients with MCD, whereas various sites, mainly in the tubulointerstitium, were stained in patients with IgAN. Additionally, each patient with both diseases showed varying degrees of staining. Based on these results, the mitochondrial injury site and degree are different for each GN, and different for each patient, even in the same GN.

Because few known prognostic factors can predict the future relapse frequency at the time of diagnosis in patients with MCD so far [7,26], the result that patients with high STING IHC staining intensity showed a higher annual relapse frequency and time-averaged proteinuria than those who did not is considered promising. Considering that in the high-intensity subgroup, the proportion of patients treated with CS and other immunosuppressants and the duration of immunosuppressant administration was longer than that in the low-intensity subgroup, the patients who showed strong STING IHC stain in kidney tissue at the time of diagnosis were less responsive to current treatment strategies.

Of the 735 patients who underwent kidney biopsy during the planned period, 43 (5.9%) and 290 (39.5%) patients were diagnosed with MCD and IgAN, respectively. At the time of study design, we planned to analyze only patients with pure MCD who had sufficient follow-up. Pathologists other than the ones who initially diagnosed the patients reviewed the pathological findings. From among the patients initially diagnosed primarily with MCD, we excluded those whose pathology showed combined diseases or those in whom MCD could not be clearly distinguished from focal segmental glomerulosclerosis. In addition, patients who were followed up for <3 years due to follow-up loss were excluded. As mentioned above, since IgAN is the most common primary GN worldwide, and the only known GN in which mitochondrial injury is related to its pathogenesis and clinical outcomes in a previous study [23], it was utilized as a control in this study. Patients with only IgAN were also enrolled through a review of the pathologic findings. After screening patients with pure IgAN, their age and sex were matched with those of MCD patients enrolled in the study to minimize the effect of baseline characteristics. Because of these strict inclusion and exclusion criteria, several patients with MCD and IgAN were excluded from the analysis; hence, the number of analyzed patients was small. The results of this study could not be generalized due to the small number of subjects; however, despite the small number of subjects analyzed, considering that there was a significant difference in the clinical outcomes according to the degree of STING IHC staining, the degree of STING IHC stain might be a factor that affects the prognosis to some extent.

The current study had several limitations. First, one of the major limitations was that we did not use kidney tissue and 24-h collected urine samples from the same patient. The COBRA cohort, a prospective GN cohort, was launched less than two years ago, so it was too short to confirm the long-term clinical outcomes of enrolled patients in this cohort. Therefore, we used prospectively collected urine samples from the COBRA cohort to measure urinary mtDNA levels only. To compare the long-term clinical outcomes according to the STING IHC staining intensity, IHC staining was performed using kidney tissues at the time of diagnosis from retrospectively selected patients with sufficient long-term clinical data for at least three years among patients diagnosed with MCD during the planned study period. Second, as mentioned above, the small sample size and the short observation period pose limitations in generalizing the results of this study to all patients with MCD and may reduce the reliability of the results. Third, there may have been a lead-time bias because the registered patients were diagnosed at different times during the disease. If the degree of mitochondrial injury was milder as the diagnosis was made early in the disease, the STING IHC staining intensity could have been weaker. Early immunosuppressant administration in these patients may affect the clinical outcomes. Considering that it is difficult to repeatedly perform a kidney biopsy, which is an invasive diagnostic method, additional experimental studies using animal models are necessary to confirm the change in mitochondrial injury degree during the natural course of MCD. Fourth, it is an important limitation that systemic mtDNA levels are not measured, as was the limitation of other GN studies using urinary mtDNA [23,27]. The possibility that urinary mtDNA increased as systemic mtDNA increased by other clinical factors was filtration into the urine cannot be excluded. In this study, age-and sex-matching was used to minimize this limitation, but systemic mtDNA level measurement would be necessary to exclude these limitations.

In conclusion, the current study results suggest that mitochondrial injury may be associated with the MCD pathogenesis, and evaluation of the signal intensity of STING IHC staining during diagnosis could be valuable in determining the prognosis of MCD. Further studies with larger cohorts and long-term follow-up durations are needed to clarify this.

## Figures and Tables

**Figure 1 jcm-11-00577-f001:**
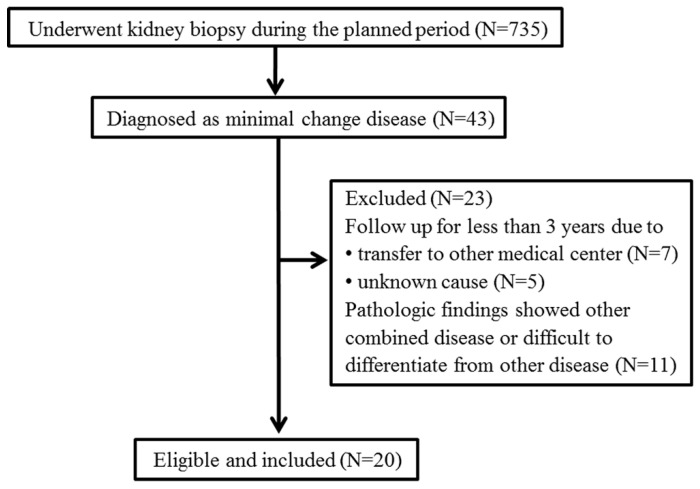
Retrospective study population.

**Figure 2 jcm-11-00577-f002:**
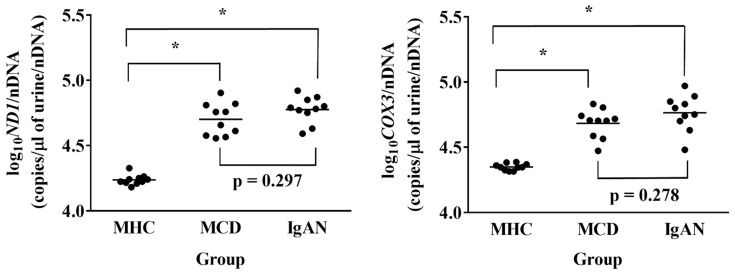
Urinary mitochondrial DNA copy numbers in the matched healthy control (MHC), minimal change disease (MCD), and immunoglobulin A nephropathy (IgAN) groups in the prospective study. Data were analyzed by via one-way ANOVA with Bonferroni correction. * *p* < 0.001. ND1, mitochondrially encoded NADH dehydrogenase 1; COX3, cytochrome-c oxidase-3.

**Figure 3 jcm-11-00577-f003:**
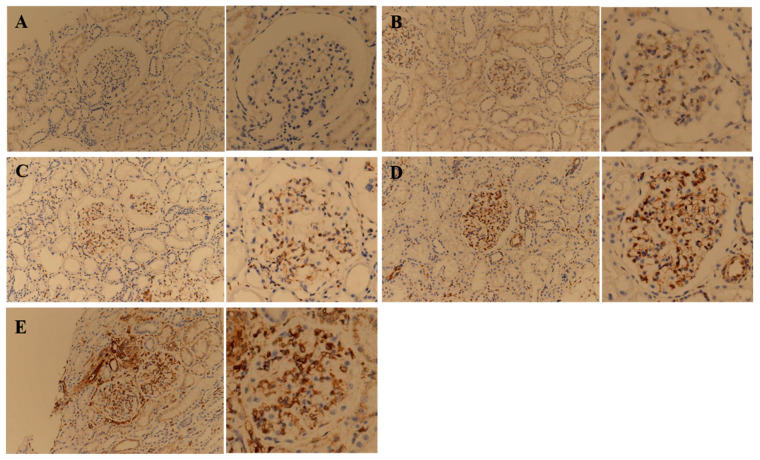
Representative images of immunohistochemical staining of STING on kidney tissue obtained from minimal change disease (MCD) and immunoglobulin A nephropathy (IgAN) patients. Figures of the same glomerulus at a higher magnification are presented to the right of each corresponding figure. MCD patients were classified into negative (**A**), 1+ (**B**), 2+ (**C**), and 3+ (**D**) according to the immunohistochemistry signal intensity. For comparison with MCD patients, an image of the STING IHC stain of the most characteristic patient with IgAN was presented (**E**).

**Table 1 jcm-11-00577-t001:** Baseline characteristics of the MHC, MCD, and IgAN groups in the prospective study.

Variable	MHC Group (*n* = 10)	MCD Group (*n* = 10)	IgAN Group (*n* = 10)	*p*-Value *
Age (years)	45.8 ± 15.2	45.8 ± 22.7	45.8 ± 16.3	>0.999
Sex (male)	4 (40.0)	4 (40.0)	4 (40.0)	>0.999
Body mass index (kg/m^2^)	22.7 ± 3.1	26.0 ± 5.4	24.5 ± 4.2	0.236
Hypertension	0 (0.0)	4 (40.0)	5 (50.0)	0.038
Systolic blood pressure (mmHg)	119.1 ± 7.9	126.9 ± 21.5	131.4 ± 19.4	0.291
Diastolic blood pressure (mmHg)	77.2 ± 7.1	76.0 ± 10.0	81.5 ± 10.6	0.396
Mean arterial pressure (mmHg)	91.2 ± 6.2	93.0 ± 12.3	98.1 ± 11.8	0.320
Baseline eGFR (mL/min/1.73 m^2^)	88.7 ± 17.6	73.0 ± 34.1	71.96 ± 18.1	0.250
Baseline proteinuria (mg/day)	80.8 ± 30.7	8659.4 ± 5559.8	902.3 ± 629.0	<0.001
Use of ARBs or ACE inhibitors	0 (0.0)	4 (40.0)	5 (50.0)	0.038

Data are presented as mean ± standard deviation for continuous variables and as number (%) for categorical variables. MHC, matched healthy control; MCD, minimal change disease; IgAN, immunoglobulin A nephropathy; eGFR, estimated glomerular filtration rate; ARBs, angiotensin II receptor blockers; ACE, angiotensin-converting enzyme. ***** One-way ANOVA or Kruskal–Wallis test for continuous variables and chi-square test or Fisher’s exact test as appropriate analysis methods for continuous and categorical variables, respectively.

**Table 2 jcm-11-00577-t002:** Site of STING immunohistochemistry stain of the MCD and IgAN groups in the retrospective study.

Stain Site	MCD Group (*n* = 20)	IgAN Group (*n* = 20)	*p*-Value
Glomerulus	13 (65.0)	13 (65.0)	>0.999
Tubule	0 (0.0)	18 (90.0)	<0.001
Interstitium	0 (0.0)	17 (85.0)	<0.001

Data are presented as the number (%) for categorical variables. MCD, minimal change disease; IgAN, immunoglobulin A nephropathy.

**Table 3 jcm-11-00577-t003:** Baseline characteristics of the high and low-intensity groups in the retrospective study.

Variable	High-Intensity Group (*n* = 6)	Low-Intensity Group (*n* = 14)	*p*-Value
Age (years)	35.3 ± 8.8	47.5 ± 19.3	0.179
Sex (male)	3 (50.0)	9 (64.3)	0.642
Body mass index (kg/m^2^)	24.9 ± 3.8	26.4 ± 3.5	0.368
Hypertension	1 (16.7)	3 (21.4)	>0.999
Systolic blood pressure (mmHg)	126.7 ± 5.2	127.7 ± 13.0	0.898
Diastolic blood pressure (mmHg)	80.0 ± 11.0	83.1 ± 7.5	0.368
Mean arterial pressure (mmHg)	95.6 ± 8.1	98.0 ± 8.6	0.639
Baseline eGFR (mL/min/1.73 m^2^)	95.2 ± 21.4	82.9 ± 19.9	0.368
Baseline proteinuria (mg/day)	6463.8 ± 5672.4	5279.3 ± 2562.9	0.966
Use of ARBs or ACE inhibitors	3 (50.0)	11 (78.6)	0.303

Data are presented as mean ± standard deviation for continuous variables and as number (%) for categorical variables. eGFR, estimated glomerular filtration rate; ARBs, angiotensin II receptor blockers; ACE, angiotensin-converting enzyme.

**Table 4 jcm-11-00577-t004:** Comparison of treatment results between the high and low-intensity groups.

Variable	High-Intensity Group (*n* = 6)	Low-Intensity Group (*n* = 14)	*p*-Value
Mean follow-up duration (years)	10.8 ± 5.4	8.7 ± 5.2	0.179
Mean annual rate of eGFR decline (mL/min/1.73 m^2^/year)	−1.2 ± 0.8	−1.4 ± 2.3	0.701
Time-averaged proteinuria (g/day)	1.18 ± 0.54	0.57 ± 0.45	0.022
Time to complete or partial remission after treatment (month)	4.4 ± 7.1	6.0 ± 7.4	0.579
Time to first relapse after complete or partial remission (month)	15.1 ± 18.8	22.1 ± 31.0	0.905
Frequency of relapses during follow-up duration (episodes/year)	0.72 ± 0.60	0.09 ± 0.22	0.022

Data are presented as the mean ± standard deviation. eGFR, estimated glomerular filtration rate.

## Data Availability

The datasets generated and/or analyzed during the current study are available from the corresponding author upon reasonable request.

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
