# Peer review of "Minimal Change Disease Is Associated with Mitochondrial Injury and STING Pathway Activation"

_jcm, 2022, doi:10.3390/jcm11030577_

Round 1

Reviewer 1 Report

     The present study evaluated the role of mitochondrial injury on the pathogenesis of minimal change disease (MCD). Additionally, authors analyzed the intensity of STING pathway activation in the renal biopsy specimen of MCD patients by the method with immunohistochemistry (IHC). Overall, the novelty of the present study is relatively high. Authors demonstrated that patients with MCD had evident mitochondrial injury in the glomerulus even though the light microscope findings show minor glomerular abnormality. Indeed, positivity of STING pathway by IHC was confirmed in epithelial side of the glomerulus in MCD, which appeared to be convincing. Furthermore, authors highlighted the significance of mitochondrial injury as a predictor or indicator for renal prognosis or response to immunosuppressive treatment, which led to increase the value of the present study. Moreover, the presentation of results and description of discussion based on the results were clearly and simply summarized. It might not to be so hard to grasp the essential points in the manuscript.

     Collectively, the present study might contain valuable and intriguing information for physicians, please consider to collect the minor points which I suggested.

Minor revision:

1, The data in patients with IgAN were utilized as positive control, is it correct? If so, please add the explanation this points in the section of material of methods. 

2, In the explanation of real-time RT PCR, it was described that authors isolated the DNA. But, I consider that not DNA but RNA was considered to be isolated for cDNA synthesis for qPCR analysis.

3, Please consider to mention whether enrolled patients with MCD for analysis of STING expressions was already administered the steroid therapy at the time of renal biopsy. I consider that treatment with steroid therapy might influence the expression of STING in the glomerulus. Please indicate the number of MCD patients who received steroid therapy before renal biopsy.

4, In Figure 3, Please add the figures of same glomerulus in more high-power field to confirm the expression of positivity in the epithelial side.

Author Response

Minor revision:

1, The data in patients with IgAN were utilized as positive control, is it correct? If so, please add the explanation this points in the section of material of methods. 

Thank you for your insightful comment. We have added an explanation of why we utilized IgAN as a control in the methods section.

“Since IgAN is the most common primary GN worldwide, and the only known GN in which mitochondrial injury is related to its pathogenesis and clinical outcomes in a previous study [23], it was utilized as a control in this study.”

2, In the explanation of real-time RT PCR, it was described that authors isolated the DNA. But, I consider that not DNA but RNA was considered to be isolated for cDNA synthesis for qPCR analysis.

Thank you for your comment. We apologize for the error. We mistakenly wrote “quantitative real-time reverse transcription-polymerase chain reaction (RT-qPCR)” in place of “quantitative real-time polymerase chain reaction (qPCR)”. Therefore, we deleted “reverse transcription-“ and “RT-“ from the methods section in the revised text.

3, Please consider to mention whether enrolled patients with MCD for analysis of STING expressions was already administered the steroid therapy at the time of renal biopsy. I consider that treatment with steroid therapy might influence the expression of STING in the glomerulus. Please indicate the number of MCD patients who received steroid therapy before renal biopsy.

Among the enrolled patients, all patients with MCD and IgAN who were treated with corticosteroids were treated after the final diagnosis via kidney biopsy. We apologize for not clarifying this detail. We have incorporated a description of this issue in the methods section.

“Among the enrolled patients, all patients with MCD and IgAN who were treated with corticosteroids were treated after the final diagnosis via kidney biopsy.”

4, In Figure 3, Please add the figures of same glomerulus in more high-power field to confirm the expression of positivity in the epithelial side.

Thank you for making this important point, which we did not consider. We have added the figures of the same glomerulus in more high-power fields as per your suggestion.

Reviewer 2 Report

  1. While comparing data between groups (1, 2 and 3) it is not appropriate to do it in following way 1 vs. 2, 2 vs. 3, and 1 vs. 3, using for example U Mann Whitney test. In such case the primary fail is multiplied by each step. I would rather recommend the use of ANOVA, incorporating all three groups and after that, in case of significant difference, the use of one post-hoc tests, e.g. Bonferonni test.
  2. I agree that the number of patients included is small and this is important limitation of the study. MCD and IgAN are one of the most predominant types of primary glomerulopathy, so in my opinion the number of patients should be enlarged.

Author Response

1. While comparing data between groups (1, 2 and 3) it is not appropriate to do it in following way 1 vs. 2, 2 vs. 3, and 1 vs. 3, using for example U Mann Whitney test. In such case the primary fail is multiplied by each step. I would rather recommend the use of ANOVA, incorporating all three groups and after that, in case of significant difference, the use of one post-hoc tests, e.g. Bonferonni test.

Thank you for your insightful comments. In accordance with your suggestion, we re-analyzed all relevant data after consulting a statistician, and were recommended the use of one-way ANOVA or Kruskal–Wallis test and chi-square test or Fisher’s exact test as appropriate analysis methods for continuous and categorical variables, respectively. We edited the figures and tables according to the results of the re-analysis. We have incorporated these in the methods and results sections.

*One-way ANOVA or Kruskal–Wallis test for continuous variables and chi-square test or Fisher’s exact test as appropriate analysis methods for continuous and categorical variables, respectively.

“More patients in the MCD and IgAN group were hypertensive and treated with ARBs or ACE inhibitors than in the MHC group.”

“The differences between two groups were compared via Mann–Whitney and Wilcoxon signed-rank tests for continuous variables and either χ2 tests or Fisher’s exact tests for categorical variables, as appropriate. One-way ANOVA or Kruskal–Wallis test for continuous variables and chi-square test or Fisher’s exact test for categorical variables, as appropriate analysis methods, were used to compare the differences between the three groups. After one-way ANOVA, Bonferroni correction was used for post-hoc analysis. “

2. I agree that the number of patients included is small and this is important limitation of the study. MCD and IgAN are one of the most predominant types of primary glomerulopathy, so in my opinion the number of patients should be enlarged.

Thank you for your comment. We agree with your opinion. As you pointed out, of the 735 patients who underwent kidney biopsy during the planned period, 43 (5.9%) and 290 (39.5%) patients were diagnosed with MCD and IgAN, respectively. At the time of study design, we planned to analyze only patients with pure MCD who had sufficient follow-up. Pathologists other than the ones who initially diagnosed the patients reviewed the pathological findings. From among the patients initially diagnosed primarily with MCD, we excluded those whose pathology showed combined diseases, including IgAN, diabetic nephropathy, and hypertensive nephrosclerosis or those in whom MCD could not be clearly distinguished from focal segmental glomerulosclerosis. In addition, patients who were followed up for <3 years due to follow-up loss were excluded. Since IgAN is the most common primary GN worldwide, and the only known GN in which mitochondrial injury is related to its pathogenesis according to the clinical outcomes of a previous study, it was utilized as a control in this study. Patients with only IgAN were also enrolled through a review of the pathologic findings. After screening patients with pure IgAN, their age and sex were matched with those of MCD patients enrolled in the study to minimize the effect of baseline characteristics. Because of these strict inclusion and exclusion criteria, several patients with MCD and IgAN were excluded from the analysis; hence, the number of analyzed patients was small. The results of this study could not be generalized due to the small number of subjects; however, despite the small number of subjects analyzed, considering that there was a significant difference in the clinical outcomes according to the degree of STING IHC staining, the degree of STING IHC stain might be a factor that affects the prognosis to some extent. We have attempted to describe the limitations of the small number of subjects in the interpretation of the results of this study in as much detail as possible in the discussion section.

“Of the 735 patients who underwent kidney biopsy during the planned period, 43 (5.9%) and 290 (39.5%) patients were diagnosed with MCD and IgAN, respectively. At the time of study design, we planned to analyze only patients with pure MCD who had sufficient follow-up. Pathologists other than the ones who initially diagnosed the patients reviewed the pathological findings. From among the patients initially diagnosed primarily with MCD, we excluded those whose pathology showed combined diseases or those in whom MCD could not be clearly distinguished from focal segmental glomerulosclerosis. In addition, patients who were followed up for <3 years due to follow-up loss were excluded. As mentioned above, since IgAN is the most common primary GN worldwide, and the only known GN in which mitochondrial injury is related to its pathogenesis and clinical outcomes in a previous study [23], it was utilized as a control in this study. Patients with only IgAN were also enrolled through a review of the pathologic findings. After screening patients with pure IgAN, their age and sex were matched with those of MCD patients enrolled in the study to minimize the effect of baseline characteristics. Because of these strict inclusion and exclusion criteria, several patients with MCD and IgAN were excluded from the analysis; hence, the number of analyzed patients was small. The results of this study could not be generalized due to the small number of subjects; however, despite the small number of subjects analyzed, considering that there was a significant difference in the clinical outcomes according to the degree of STING IHC staining, the degree of STING IHC stain might be a factor that affects the prognosis to some extent.”

“Second, as mentioned above, the small sample size and the short observation period pose limitations in generalizing the results of this study to all patients with MCD and may reduce the reliability of the results.”

“Further studies with larger cohorts and long-term follow-up durations are needed to clarify this.”

Round 2

Reviewer 2 Report

I have no further comments.